# Three-Dimensional Modeling of CpG DNA Binding with Matrix Lumican Shows Leucine-Rich Repeat Motif Involvement as in TLR9-CpG DNA Interactions

**DOI:** 10.3390/ijms241914990

**Published:** 2023-10-08

**Authors:** Tansol Choi, George Maiti, Shukti Chakravarti

**Affiliations:** 1Department of Ophthalmology, NYU Grossman School of Medicine, New York, NY 10016, USA; tc3582@nyu.edu; 2Department of Pathology, NYU Grossman School of Medicine, New York, NY 10016, USA

**Keywords:** lumican, TLR9, CpG DNA, extracellular matrix, LRR motifs, protein-DNA docking, inflammation, collagen

## Abstract

Lumican is an extracellular matrix proteoglycan known to regulate toll-like receptor (TLR) signaling in innate immune cells. In experimental settings, lumican suppresses TLR9 signaling by binding to and sequestering its synthetic ligand, CpG-DNA, in non-signal permissive endosomes. However, the molecular details of lumican interactions with CpG-DNA are obscure. Here, the 3-D structure of the 22 base-long CpG-DNA (CpG ODN_2395) bound to lumican or TLR9 were modeled using homology modeling and docking methods. Some of the TLR9-CpG ODN_2395 features predicted by our model are consistent with the previously reported TLR9-CpG DNA crystal structure, substantiating our current analysis. Our modeling indicated a smaller buried surface area for lumican-CpG ODN_2395 (1803 Å^2^) compared to that of TLR9-CpG ODN_2395 (2094 Å^2^), implying a potentially lower binding strength for lumican and CpG-DNA than TLR9 and CpG-DNA. The docking analysis identified 32 amino acids in lumican LRR1–11 interacting with CpG ODN_2395, primarily through hydrogen bonding, salt-bridges, and hydrophobic interactions. Our study provides molecular insights into lumican and CpG-DNA interactions that may lead to molecular targets for modulating TLR9-mediated inflammation and autoimmunity.

## 1. Introduction

Lumican is a member of the small leucine-rich repeat proteoglycan (SLRP) family, widely present in various extracellular matrix (ECM) types [1,2]. A few SLRPs are post-translationally modified with glycosaminoglycan side chains in mature ECMs. Lumican is a modified keratan sulfate (KS) proteoglycan in connective tissues like the tendon, cornea, and cartilage, but it is secreted as a simple glycoprotein by activated fibroblasts during infections and inflammation [3,4,5]. Many of the core proteins consist of tandem repeats of leucine-rich repeat (LRR) motifs. A body of work, including transmission electron microscopy of collagen-rich tissues, recombinant core proteins, and biochemical approaches show that lumican and other SLRP members associate with collagen fibrils to restrict their lateral growth and assembly, and that they localize to different bands on the fibrils [6,7,8,9,10]. Functionally, the SLRPs regulate collagen fibril growth and proper assembly of ECMs, such that SLRP-null mice develop abnormal connective tissues [11,12,13,14]. SLRP interactions with collagen fibrils engage multiple LRR repeats in the core proteins [15,16]. In addition to collagen-fibril regulations, emerging studies indicate that SLRPs interact with cell surface receptors, chemokines, and pathogen-associated molecular patterns, cementing their significant involvement with host immune responses [2,4]. Given that the SLRP core proteins are largely comprised of LRR motifs, they are members of the LRR superfamily of proteins, which includes pathogen recognition toll-like receptors (TLRs).

Host pathogen recognition receptors (PRR) include TLRs that mediate the host response to microbial-pathogen-associated molecular patterns, as well as certain host-derived-danger-associated molecular patterns [17,18]. Human and mice have 10 and 12 TLRs, respectively. TLRs 1, 2, 4, and 5 localize to the plasma membrane, whereas TLR3, 7, 8, and 9 are destined to endosomal compartments for PAMP recognition and signaling [18]. TLR9 recognizes viral double stranded (ds) DNA in acidic endolysosomal compartments. TLR9 may also interact with host nuclear DNA as well as mitochondrial DNA released from dying cells during infections and inflammation, which contributes to an increased risk for autoimmunity [19]. Adequate recognition of pathogenic DNA, but minimal recognition of self-DNA, is achieved through guided trafficking and cleavage of TLR9 in endolysosomes. Our experimental study shows that ECM proteins like lumican may modulate the recognition of pathogenic DNA [5]. Future studies will also address whether ECM proteins minimize recognition of host-derived DNA by TLR9. Lumican and TLR9 have multiple tandem repeats of LRR motifs. LRR motifs from TLR9 have already been shown to interact with some synthetic forms of DNA ligands [20,21]. Our 3D modeling of lumican binding to DNA ligands, for the first time, identifies candidate DNA-binding LRR motifs in lumican.

LRR motifs, first discovered in 1985, are approximately 24-amino-acid-long highly conserved leucine-rich stretches, with regularly spaced hydrophobic residues [22]. Over 6000 proteins having LRR motifs are listed in PFAM, PRINTS, SMART, and InterPro databases [23]. These LRR domains have a highly conserved segment (HCS) and a variable segment (VS) [24,25]. The HCS, LxxLxLxxNxL, contains conserved leucine residues (Leu), which may be substituted with Isoleucine (lle), phenylalanine (Phe), or valine (Val). The ninth residue is usually an asparagine (Asn), serine (Ser), threonine (Thr), or cysteine (Cys). LRR superfamily proteins generally have a horseshoe shape, where the Leu, Ile, Phe, and Val at position 1, 4, 6, and 11 of HCS form the hydrophobic core of the protein comprised of beta sheets [24]. The Asn at position 9 connects LRR motifs to each other by forming a hydrogen bond. The VS part forms loop and helix conformation at the outer convex surface [24]. 

In addition to the shared structural similarities discussed above, the SLRP core proteins harbor some TLR-like functions and interact with pathogen recognition signals. In a recent study, we found that macrophages from lumican-null mice show increased innate immune responses to CpG DNA, a ligand for endosomal TLR9, while the addition of recombinant lumican suppressed TLR9 response [5]. The study further demonstrated that exogenous lumican and CpG DNA are co-localized in early endosomes, which are not enriched for TLR9. An in vitro recombinant lumican binds CpG DNA in a dose-dependent manner. The data collectively indicated that lumican binds CpG DNA and sequesters it in TLR9-poor endosomes to restrict responses to DNA. However, the molecular underpinnings in the lumican-DNA interactions are unknown. 

Here, we used in-silico approaches to predict the LRR sequences in lumican and their mode of interactions with CpG DNA. We modeled lumican-CpG DNA interactions using the 22-mer synthetic CpG DNA, henceforth termed as CpG ODN_2395, which has been shown to experimentally stimulate mouse and human TLR9 responses. A few in silico studies have examined TLR9 interactions with CpG DNA forms other than the CpG ODN_2395 we studied here [21,26]. Prior crystallographic studies on the TLR9 extracellular domain (TLR9 ECD) used a variation of the 12-mer synthetic CpG ODN_1668, CATGACGTTCCT, and a 6-mer that binds to a second DNA binding region to enhance signals in human, equine, and bovine TLR9 [27]. However, there are no experimental or model predictions of TLR9 interactions with the 22-mer CpG ODN_2395 ligand. Therefore, to ascertain the extent to which lumican mimics TLR9 in its interactions with the CpG ODN_2395 ligand, we also modeled mouse TLR9 ECD interactions with this ligand. Our studies provide new insights into matrix lumican and mouse TLR9 interactions with a CpG ODN_2395 known to evoke a strong innate immune response in vivo.

## 2. Results

### 2.1. 3D Modeling of Lumican and LRR Motif-Carrying Domains in Lumican and TLR9

Because mouse lumican has not been crystallized, we used the AlphaFold Protein Structure Data Base [28,29] to visualize the predicted structure of mouse lumican in a PDB format. We also superimposed TLR9 LRR1–13 on the lumican model for visual comparison (Figure 1A). The analysis range of lumican (amino acid residues 40–309) yielded a high per residue confidence metric or predicted local distance difference test (pLDDT) score of more than 90%, and a predicted aligned error (PAE) ranging from 0 to 15 Å.

The LRR motifs, based on their consensus sequence, can be categorized into seven classes: bacterial (S), cysteine containing (CC), plant specific (PS), SDS22, ribonuclease inhibitor like (RI), irregular (I), and typical (T) [25]. In a pairwise alignment using Clustal Omega, lumican shared an overall identity of 22% and a similarity of ~40% with TLR9 (Appendix A). We detected the highest similarity between lumican LRR motifs 1–11 and LRR 1–13 of mouse TLR9 (Figure 1A). Lumican follows a typical pattern of LRR repeat consensus, with approximately 20 residues per repeat, where the central ten LRR motifs (LRR1–11) are a mixture of T, S, SDS22, and PS classes, flanked by an N-terminal LRRNT motif with 4 cysteine residues and a C-terminal LRRCT motif with 2 cysteine residues (Appendix A). TLR9 has a total of 26 LRR motifs, where several are irregular, with insertions of variable lengths of amino acid that disrupt the average 26 residues per LRR structure. Thus, LRR 2, 5, 8, and 11 of mouse TLR9 have 10–16 insertions at position 10 after the 9th consensus Asn residue [30]. Surface charge distribution analysis along the concave surface of lumican shows that it consists of basic and acidic patches, and involves an equal number of basic and acidic interface residues (Figure 1B and Appendix A). On the other hand, in TLR9, LRR1–13 and LRR13–26 motifs along the concave surface are predominantly basic patches (Figure 1C and Appendix A).

### 2.2. CpG ODN_2395 Docking and Interaction Surfaces on Lumican and TLR9 ECD

As TLR9 interactions with CpG ODN_2395 has not been studied before, we evaluated the docking of CpG ODN_2395 on the “unliganded form” (PDB: 3WPF) of mouse TLR9. Our docking analysis of CpG ODN_2395 on lumican identified LRR 1–11 (Table 1) and on LRR2–8, 13, 15, and 20–25 on TLR9-ECD (Table 2) as the most likely sites of interactions. The reasons for opting for a rigid-docking model is provided in the Section 4. Briefly, TLR9 has been crystalized with the 12 mer shorter ligand, CpG ODN_1668, and has been reported (PDB: 3WPC). The longer CpG ODN_2395 has only 6 aligned sequences in common with the shorter ligand, and therefore, HDOCK rejected the template with sequence coverage < 50% from the candidate templates. Instead, it selected TLR3 complexed with the double stranded (ds) lncRNA Rmrp (PDB: 7DA7) as a common template for lumican and TLR9 docking with CpG ODN_2395 that gave sequence coverage greater than 50% for both the ligand (CpG ODN_2395) and the receptors (lumican and TLR9). While this is interesting, it is not related to the biologically relevant question of how lumican interacts with the longer CpG ODN_2395 to modulate the TLR9 response in mouse models and cell culture systems.

We next visualized CpG ODN_2395 interactions with lumican and TLR9 using 3D modeling. A large area along the concave surface of lumican engages with CpG ODN_2395, where LRR2–8 from the N-terminal end forms a binding groove for the 5′ end of CpG ODN_2395 (Figure 2A,B).

We also visualized the CpG ODN_2395 interacting surfaces in TLR9 ECD. The binding region includes LRR2–13 from the N-terminus and LRR15, 20–25 from the C-terminus (Figure 2C,D and Table 2). This is different from those reported earlier for interactions of TLR9 ECD with the shorter CpG ODN_1668, where only the N-terminal, and C-terminus LRR20–22 are engaged (Appendix A). We further examined the predicted buried surface area (BSA), defined as the surface area buried or unavailable to solvents in protein bound to CpG DNA. Analysis of the lumican—CpG ODN_2395 interaction yielded a BSA of 1803 Å^2^ (Figure 2A), whereas the TLR9 ECD-CpG ODN_2395 interaction yielded a BSA of 2094 Å^2^ (Figure 2B). This indicates that compared to lumican, a slightly larger surface area of TLR9 engages with CpG ODN_2395. By contrast, a previous study on TLR9 ECD interactions with the smaller 12 mer CpG ODN_1668 showed 1430 Å^2^ BSA, predictably suggesting smaller surface occupancy by the smaller CpG DNA [21].

### 2.3. Lumican and CpG ODN_2395 Interactions Involve Hydrogen Bonding, Salt-Bridges and Hydrophobic Interactions

The lumican-CpG ODN_2395 interface involves 32 amino acid residues from lumican, and 9 nucleotide bases from the CpG ODN_2395, at an average resolution of 3.3 Å. The interacting atoms from the bases in CpG ODN_2395 and the amino acids from lumican are shown (Figure 3A and Table 3). The interacting residues include 7 basic (Lys119, His98, His121, His237, Arg73, Arg234, Arg309) residues, two hydrophilic asparagine residues (Asn74, Asn123), seven acidic residues (Glu257, Glu281, Asp97, Asp140, Asp189, Asp212, Asp259), and 16 other hydrophobic and polar uncharged amino acid residues. The interactions include 11 hydrogen bonding interactions followed by 4 salt-bridge and 2 hydrophobic interactions (Figure 3A and Table 3). Hydrogen bonding interactions with CpG ODN_2395 pyrimidine/purine bases are predominant, while 82% H-bond donor residues reside in lumican, and 82% of H-bond acceptors reside in CpG ODN_2395 (Table 3). Biochemically, the amino acids in lumican that form the 11 H-bonds include Asn74, Arg234, and Arg309, two acidic Asp259 and Glu281, a neutral Ser261, two acidic Gln142, Gln166, and three aromatic Tyr164, Tyr187 and Tyr210. The four salt bridges are formed between opposite charges in polar/charged residues, including Glu257, Asp189, His98, and His121. Acidic residues Glu257 and Asp189 form salt bridges with the G3 and G6 purine bases, respectively, of CpG ODN_2395 (Figure 3B–D). However, the positive charge centers of the bases form electrostatic interactions with the Glu257, and negative charges of the carboxyl side chains form interactions with the Asp189 (Figure 3D). The positive imidazole side chain of His98 and 121 forms salt-bridges with the ribose-phosphate backbones of CpG ODN_2395. Hydrophobic/pi-stacking interactions are observed between the aromatic ring of Tyr 262 and the pyrimidine base ring of T1, where the Tyr aromatic ring is perpendicular to the thymidine base ring, making it ideal for pi-stacking interactions (Figure 3C).

The hydrophobic and the basic residues of lumican form the interface with the first and third groove of CpG ODN_2395. The lumican residues that form close contact with the first groove of CpG ODN_2395 include Asp189, Asp259, Glu257, Glu28, Arg234, Arg309 and Tyr164. While Tyr187, Tyr210 and Ser261 form salt bridges/H-bonds with the two 5′ CpG repeats (5′-TCGTCG). These “hot spot” lumican residues in LRR5–11 form a groove that interacts with the two 5′ CpG repeats. These lumican residues are therefore somewhat biochemically similar to the TLR9 ECD LRR2–13, as predicted by the pairwise alignment score (Appendix A).

### 2.4. TLR9 ECD Interactions with CpG ODN_2395 Involve N-Terminal LRR2–13 Motifs

The TLR9-CpG ODN_2395 interface encompasses 33 residues of TLR9, and 10 nucleic acids of CpG ODN_2395 at an average resolution of 3.5 Å (Table 2, Figure 2C,D). As indicated by the APBS map, the interface residues involve many basic residues that form distinct basic patches in the N and C terminal side of TLR9 (Appendix A). The N-terminal LRR2–13 of TLR9 contributes the most towards the binding affinity, comprising 553 Å^2^ out of 878 Å^2^ of the buried surface area of TLR9 (Figure 2C). The TLR9 N-terminal LRR2–13 region contains seven basic residues: Lys95, Asn129, His152, Lys181, His203, Lys207, Asn263, and one acidic residue, Asp175 (Appendix A). There are 15 additional neutral, polar, or hydrophobic (Ser131, Ser149, Ser151, Ser205, Tyr132, Tyr179–180, Tyr208, Tyr224, Val171, Phe173, Leu226, Pro262, Gln399 and Met400) residues in these interacting LRRs (Table 2). However, the C-terminal LRR15–25 contains six basic residues: Arg482, Arg613, Asn640, His642, Arg662, and His735 (Appendix A). Our results show that the predominant type of interaction for TLR9-CpG ODN_2395 is hydrogen bonding, involving 12 H-bond interactions where 75% of the H-bond donors reside in TLR9 and 75% of the acceptors in CpG ODN_2395. Specifically, the H-bond donors are four tyrosine (Tyr132, Tyr179, Tyr180, and Tyr665) and other residues (His203, Arg613, Asn263, Ser131, Ser205, and Lys207) (Figure 4A–E). The salt bridge interactions are the next prevailing interaction, mostly involving charge–charge interactions between positively charged basic residues (Lys95, His152, Arg482, Arg613, His642, and His735) and negatively charged sugar-phosphate backbone atoms (Figure 4A and Table 4). One exception was the acidic residue Asp175, where the carboxylic side chain forms a salt bridge with the positive charge center of the G3 purine base ring (Figure 4B,C). Overall, 11 out of the 18 bonding interactions, including H-bond/salt-bridges, are from LRR2–13. The N-terminus LRR2–13 of the TLR9-CpG ODN_2395 interface comprises 63% of BSA (878 Å^2^), while LRR15–26 from the C-terminus makes up only 37% of BSA (Table 2). Thus, the C-terminus of LRR15–26 has low binding affinity for CpG ODN_2395, which is consistent with experimental data that shows its reduced involvement in TLR9 downstream signals [20]. The His152, His203, Tyr132, Tyr179, Tyr180, Ser131, Ser205, Lys207, and Asn263 of LRR2–8 interacts closely with the 5′ end hexamer (5′-TCGTCG) of CpG ODN_2395, forming the H-bonds and salt bridges (Figure 4B,C). Additional interacting atoms from CpG ODN_2395 are also shown (Figure 4D,E and Table 4).

As indicated previously, TLR9 interacting residues with CpG ODN_1668 12 mer are clustered in LRR1–8 from the N-terminus, which includes the generic PAMP binding surface LRR2, 5, and 8, followed by LRR20–22 from the C-terminus [21]. The TLR9-CpG ODN_1668 12 mer interaction region includes N-terminus basic patches of TLR9 that engage in hydrogen bonding and satisfies charge complementarity to the negatively charged ribose-phosphate backbone (Appendix A). The prevailing interactions between LRR1–8 and ODN_1668 are H-bonding between sugar-phosphate backbones.

### 2.5. The Predicted Effects of Collagen Type I on Lumican Interactions with CpG ODN_2395

Multiple studies have shown that lumican associations with collagen type I regulate collagen fibrillogenesis in vitro [7], and that similar interactions in vivo impact collagen fibril assembly [12,15,31]. One study found that synthetic peptides from the lumican LRR5–7 region competes with full length lumican and significantly compromises collagen type 1 fibrillogenesis [9]. Moreover, the D212N recombinant lumican, which carries this mutation in LRR5–7, disrupts both the binding of lumican to collagen and subsequent collagen fibrillogenesis [9]. Our simulation shows that lumican, upon binding to CpG ODN_2395, creates a significant amount of BSA (760 Å^2^ of 1803 Å^2^, which is 42%). The BSA of most bona fide protein-ligand interactions are observed within this range [32]. Our prior interface analysis identified Asp212, Gln166, Tyr210, Asp189, Tyr164, Thr208, Tyr187, and Phe162 as interacting with CpG ODN_2395, and these fall within the LRR5–7 that interact with collagen (Figure 5A). The Asp212 (D212) residue within LRR7, which is involved in fibrillogenesis, is predicted to be buried upon CpG binding. This site is also highly conserved in lumican from different species as shown by multiple sequence alignment, indicating its functional importance (Figure 5B). These analyses also imply that collagen binding will likely interfere with the ability of LRR5–7 to bind CpG-DNA.

## 3. Discussion

Our earlier experimental studies showed that lumican binds CpG ODN_2395, a synthetic ligand for TLR9, to restrict response and subsequent induction of proinflammatory cytokines by TLR9 [5]. Here, for the first time, we used in silico approaches to identify the DNA-interacting region within the core protein of lumican at a molecular level, and characterize the types of bonding involved in these interactions. Like other LRR family members, the lumican core protein is made up almost entirely of tandem repeats of LRR motifs. Our analysis shows that while overall identity (22%) and similarity (~40%) between mouse lumican and TLR9 were not high, the lumican LRR motifs, except its last LRRCE motif, share the strongest similarity with the anterior half of TLR9—namely LRR 1–13. Like other LRR proteins, lumican and TLR9 have a solenoid shape, where highly conserved residues are mainly located in the concave surfaces. Interactions with CpG DNA largely occur through the concave surface of the solenoid structures. Both proteins have a mixture of T, S, SDS22, and PS classes of LRRs, where S and T are considered to interact with proteins and DNA, and also have role in immunity and defense. Indeed, lumican LRR–1, 2, and 5–7, which are a mixture of S and T class, are predicted to engage with CpG ODN_2395.

Analysis of BSA in the interactions of lumican (1803 Å^2^) or TLR9 ECD (2094 Å^2^) with CpG ODN_2395 suggest that, compared to TLR9 ECD, a slightly lower surface area of lumican is involved. This may imply a weaker binding strength between lumican and CpG ODN_2395 than between TLR9 ECD and CpG ODN_2395. However, from a functional standpoint in tissues, ECM proteins like lumican are present at a much greater abundance than pathogen recognition receptors like TLR9, and are thus expected to be impactful. Multiple, still incompletely understood mechanisms, are in place to control the TLR9 response to pathogenic DNA as well as self-DNA for controlled inflammatory responses to infections, and minimize harmful responses to self-DNA [18,19,33,34,35,36]. Synthesized in the endoplasmic reticulum, TLR9 is bound by Unc93B1, a membrane adaptor protein, which facilitates proper folding and incorporation into transport vesicles that apparently takes it to the plasma membrane first, before its final location in acidic endolysosomes where it is proteolytically cleaved to release a smaller functional protein [37]. Our prior studies show that there is an abundance of lumican at the cell surface, and it is also endocytosed by immune cells [5]. Therefore, binding of DNA ligands by lumican at the cell surface can counteract potential ligand interactions with TLR9 transiently located at the cell surface. We also showed that endocytosed lumican colocalizes with CpG DNA, and these endosomal sites are low in TLR9. This is another way that ligand interactions with TLR9 are reduced by lumican. We still do not know how lumican favors the separation of the DNA ligand from TLR9 in endosomes. The overall effects of lumican, and likely some of the other SLRPs, are to reduce ligand availability and interactions with TLR9, as well as to reduce inflammatory responses to pathogenic and self-DNA.

A potential limitation of our study is that we used the lumican core protein instead of the glycoprotein or the proteoglycan forms of lumican in our 3D modeling. The negatively charged KS side chains of lumican will impact interactions of the core protein. However, we have shown experimentally that during infections and inflammation, much of the lumican interacting with macrophages is not the proteoglycan form, but the simple glycoprotein form without the KS side chains [5]. 

Another important limitation of our study is the use of the rigid, B-form of the CpG ODN_2395 in predicting its interactions with lumican and TLR9. The electrostatic repulsions between the highly negatively charged phosphate groups lead to conforma-tional flexibility in ssDNA posing a challenge in modeling the ssDNA-protein interac-tions. Earlier studies have used small-angle x-ray scattering (SAXS) to probe the dy-namic and flexible conformations of ssDNA in solution of varying ionic strength [38,39,40]. Sim et al, have used SAXS to study the flexibility of ssDNA under different salt concentration [41]. In another study the interactions between the eukaryotic replication protein A (RPA) and an 8 mer or 14 mer ssDNA was studied by combining SAXS data to generate numerous dynamic conformation with molecular dynamics (MD) simulations that may predict the dynamic interactions more accurately [42]. Few MD simulations studies have been instrumental in understanding the flexible interactions of ssDNA with ssDNA-binding proteins (SSB) [43,44] and the K homology domains (KH3 and KH4) [45]. MD simulations are highly accurate in probing the atomic mo-tions and structural flexibility of dsDNA, but often less effective in case of ssDNA [46,47]. MD simulations were also used to study the conformational changes of TLR7 and TLR8 and their agonists, where the RMSD values of TLR7 and 8 backbone atoms fluctuate from their mean position within a scope of 2–4 Å after 30 ns simulation [48]. Therefore, under physiological solutions the interacting atoms between CpG ODN_2395 and lumican and TLR9 could be different from our results generated using the rigid B form CpG DNA model. The flexible nature of CpG ODN_2395 can lead to an ensemble of major and minor conformational derivatives with lumican and TLR9 that might yield mean interacting amino acids and nucleotides that could be different from this limited study. To further validate our study, MD simulations like AMBER [49] that calculate the force of each atom as a function of their positions can be used over 100 ns period for CpG ODN_2395, lumican and TLR9. Applying similar ap-proaches to study CpG DNA – lumican or TLR9 complexes is also challenging due to few available structures of CpG DNA in PDB database and low sequence identity, making it difficult to develop template-based models. Despite these limitations, this study is the first in silico attempt to predict the amino acid residues of lumican that might be interacting with CpG DNA, corroborating our previous experimental finding that lumican binds to CpG ODN_2395 in solid-phase binding assay [5].

In summary, our study provides molecular insights into the interactions between lumican and a synthetic ligand for TLR9, providing an avenue for the future development of drug targets. Additional studies are required to expand our knowledge of the full extent of natural TLR9 ligands that lumican and other SLRPs interact with to regulate inflammation and autoimmunity.

## 4. Materials and Methods

### 4.1. Data Sources

FASTA sequences of mouse TLR9 and lumican were obtained from UniProt with primary accession numbers Q9EQU3 and P51885, respectively. The Class C CpG ODN_2395 sequence (5′-TCGTCGTTTTCGGCGCGCGCCG-3′) was obtained from Invivogen (#tlrl-2395). We used the crystal structure of the mouse TLR9 extracellular domain (TLR9 ECD) not bound to a ligand “unliganded form” (PDB:3WPF) obtained from Protein Data Bank [50]. Because mouse lumican has not been crystallized, we used the AlphaFold Protein Structure Database [28,29] to visualize the predicted structure of mouse lumican in a PDB format. AlphaFold-predicted structures are highly accurate, with a median backbone accuracy of 2.8 Å r.m.s.d._95_ (95% confidence interval = 2.7–4.0 Å), and produces highly accurate side chains. The all-atom accuracy of AlphaFold is 1.5 Å r.m.s.d._95_ (95% confidence interval = 1.2–1.6 Å) compared to other currently used methods. AlphaFold generates per-residue confidence metrics called the predicted local distance difference test (pLDDT) score on a scale of 0–100. Structures with pLDDT > 90 are expected to be modelled with high accuracy [28]. The analysis range of lumican (amino acid residues 40–309) yielded a high pLDDT score of more than 90%, and a predicted aligned error (PAE) ranging from 0 to 15 Å.

### 4.2. TLR9 and Lumican Local Similarity Analysis

Lumican and TLR9 FASTA sequences from UniProt were submitted to Clustal Omega local alignment tools [51] to align the most similar regions between two sequences. The lumican sequence 3–321—corresponding to LRR1–11 out of 338—and TLR9 sequence 19–408—corresponding to LRR1–13 out of 1032—were aligned. We introduced 121 gaps to align both sequences with a default gap opening penalty score of 10 and extension penalty score of 0.5. The matched sequences have 165 conserved and semi-conserved residues out of 415, showing 22% sequence identity and 40% sequence similarity, with an alignment score of 225.5 after subtracting the gap opening and extension penalty score.

The results are the following:Aligned_sequences: LUM_MOUSE and TLR9_MOUSEGap_penalty: 10.0Extend_penalty: 0.5Length: 415Identity: 92/415 (22.2%)Similarity: 165/415 (39.8%)Gaps: 121/415 (29.2%)Alignment Score: 225.5

The local structural alignment between two proteins were conducted using the PyMOL Align command line accessed on 25 June 2023 (https://pymolwiki.org/index.php/Align) [52]. It performs a local sequence alignment between two proteins using a Clustal Omega local sequence alignment tool, followed by structural superimposition with the cycles of refinement in order to reject structural outliers. The lumican LRR1–11 and TLR9 LRR1–13, totaling 1494 atoms, were aligned with five cycles of structural refinement—the average distance between the atoms of two superimposed proteins, or an RMSD value 6.4 Å. Structural outliers with the average RMSD = 6.83 Å were rejected during the cycles (53 atoms).

The following parameters were used to run PyMOL structural alignment between Lumican and TLR9:outlier rejection cycles: 5RMS outlier rejection cutoff: 2.0Mobile_state: 0 = all statesTarget_state: 0 = all statesTransform: 1 = superposition

The results obtained are the following:Alignment score: 225.5ExecutiveAlign: 1819 atoms aligned.ExecutiveRMS: 53 atoms rejected during cycle 5 (RMSD = 6.83 Å)Executive: RMSD = 6.411 Å (1494 to 1494 atoms)

### 4.3. Surface Charge Distribution Analysis

The PyMOL Plugin APBS (Adaptive Poisson-Boltzmann solver) electrostatics software version 2.1 allows visualization of the distribution of acidic (negatively charged) and basic (positively charged) amino acids on the surface of lumican and TLR9 3D structures, but does not provide the deprotonation/protonation status of individual amino acid residues. The preparation of molecules, including adding missing molecules and assigning atomic charges, was automated by pdb2pqr, and grid spacing for APBS calculation was 0.50 Å, which are the default settings. The following parameters were used to run APBS electrostatics to calculate the surface charge of lumican and TLR9:Prepare molecule with method: pdb2pqrCalculate map with APBS with grid spacing: 0.50 ÅMolecular surface visualization with electrostatic potential range (+/−): 5.00

### 4.4. Molecular Modeling of CpG ODN_2395

The model structure of Class C CpG ODN_2395 was built using BIOVIA Discovery Studio (version 4.5) biopolymer tools. Given that TLR9 demonstrated strong binding and immunostimulatory activity with single stranded CpG DNA, we built a single stranded CpG ODN_2395 as a B-form using standard helix settings (right-handed, 10 bp per turn, 3.4 Å vertical rise per bp, helical diameter 19 Å) without minimizing the structure.

### 4.5. Rigid Docking Prediction

We used the HDOCK server for protein-ssDNA interaction. The HDOCK server searches for sequence similarities for the query receptor and ligand against the PDB database for homology modeling and homologous template searching, followed by the prediction of putative binding sites using the HDOCKlite global docking algorithm version 1.1 that uses the homologous template for accurate docking prediction [53]. The HDOCK software version 1.1 chooses a template based on maximum sequence coverage and maximum sequence identity for both ligand and receptor, and the threshold for sequence coverage is 50%. Although the TLR9 crystallized structure in complex with CpG ODN_1668 12 mer exists in the PDB database (PDB: 3WPC), the CpG DNA sequence coverage was 27% (6/22), which was below the HDOCK threshold of 50%. Therefore, HDOCK automatically excludes the templates with sequence coverages less than 50% of the template library. Instead, HDOCK chose TLR3 complexed with the double stranded lncRNA (PDB: 7DA7) as a modeling template that gave a sequence coverage greater than 50% for both the ligand (CpG ODN_2395) and receptors (lumican and TLR9), as depicted in the table below (Table 5).

The docking prediction was conducted with the following settings:ligand type: ssDNAreceptor/ligand format: PDB file.

Docking type: global docking (default)

The interface residues were determined within a distance of 5.0 Å between the receptor and the ligand. The prediction generated 100 complexes. In this study, only top1 with the highest confidence and lowest docking score was considered for the analysis.

### 4.6. Protein-DNA Interface Analysis

The protein-DNA complex and residues at the interface were obtained from the HDOCK server and visualized in PyMOL. The interface area of the protein-CpG DNA complex was determined by the buried surface area (BSA), which is the sum of the accessible surface area (ASA) of unliganded protein and free CpG DNA, subtracted from that of the protein-CpG DNA complex. The buried surface area of each individual residue was calculated from the ASA of the unbound state of the individual residue subtracted from the ASA of the CpG ODN-bound state of the individual residue. The buried interfacial surface area is directly proportional to the affinity; therefore, with an increase in BSA, the binding affinity between the receptor and ligand increases [54]. The buried surface area of the protein, CpG DNA, and protein-CpG DNA complex was calculated using the PyMOL built-in command line Get_area. The probe diameter was set to 1.4 Å [26], which is the diameter of water molecule.

The following equation was used to evaluate the BSA and loss of accessible surface area of individual residue:Total buried surface area (Å^2^) = (accessible surface area_protein_ + accessible aurface area_CpG_DNA_) − (accessible surface area_protein-CpG DNA_)Buried surface area of individual residue (%) = (accessible surface area_Free_residue_ − accessible surface area_Bound_residue_)/accessible surface area_Free_residue_

The hydrogen bonding, salt-bridges, and hydrophobic interactions in the protein-DNA interface were analyzed using the Protein-Ligand Interaction Profiler (PLIP) [55]. PLIP uses four steps of algorithms to detect interaction patterns: structure preparation to add missing atoms, functional characterization to detect hydrogen bonding acceptor/donor atoms and hydrophobic molecules, rule-based matching based on geometric criteria, and filtering of redundant interactions. The complex generated from the HDOCK server was submitted to PLIP as a PDB format, and the interaction data were visualized in PyMOL.

## 5. Conclusions

Our results indicate that 32 amino acids within the concave surface of lumican primarily interact with CpG ODN_2395. The residues encompassing LRR2–8 form a binding groove for the 5′ end of CpG ODN_2395. The lumican-CpG ODN_2395 interactions are mainly through hydrogen bonding, salt-bridges, and hydrophobic interactions. Lumican shares an overall identity of 22% with TLR9 ECD, which is the specific pathogen recognition receptor for CpG ODN_2395. TLR9 ECD comprises LRR1–26, with 33 amino acids interacting with CpG ODN_2395 through hydrogen bonding and salt-bridges. Our docking analysis demonstrated a smaller buried surface area for lumican-CpG ODN_2395 (1803 Å^2^) compared to that of TLR9-CpG ODN_2395 (2094 Å^2^), which may imply a lower binding strength for lumican with CpG-DNA than that of TLR9 with CpG-DNA.

## Figures and Tables

**Figure 1 ijms-24-14990-f001:**
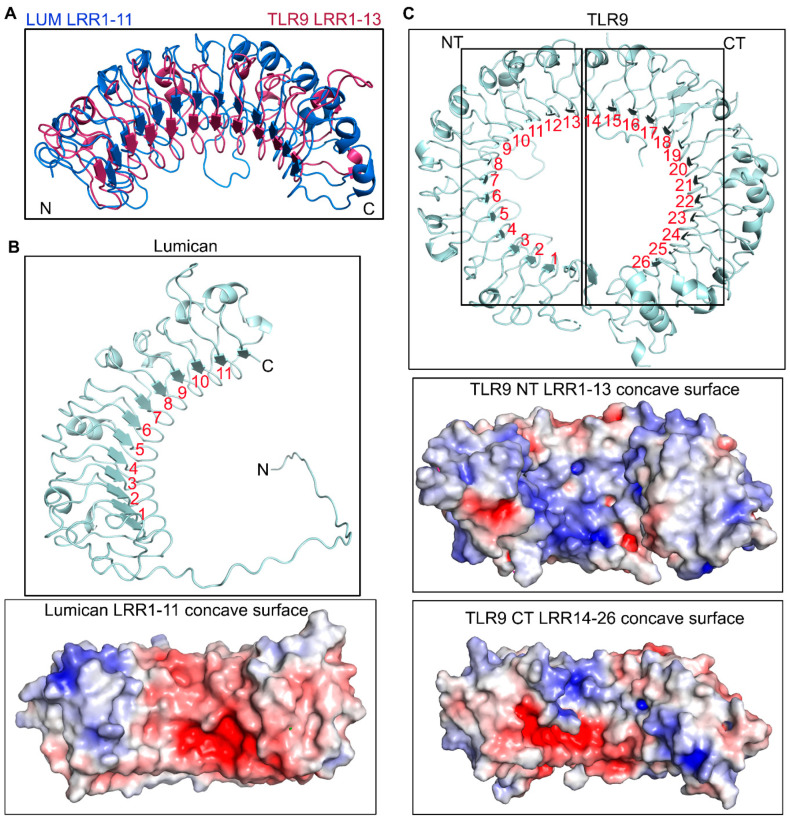
Lumican model structure and similarities with TLR9. (**A**) The model depicts the structural alignment of TLR9 LRR1–13 and lumican LRR1–11 based on ClustalW pairwise sequence alignment. (**B**,**C**) Horseshoe-like structure model and surface charge representation showing basic (blue) and acidic (red) properties of the lumican concave surface showing LRR1–11 repeats (**B**), and TLR9 showing LRR1–26 repeats (**C**). N or NT indicates N-terminal end and C or CT indicates C-terminal end of the protein.

**Figure 2 ijms-24-14990-f002:**
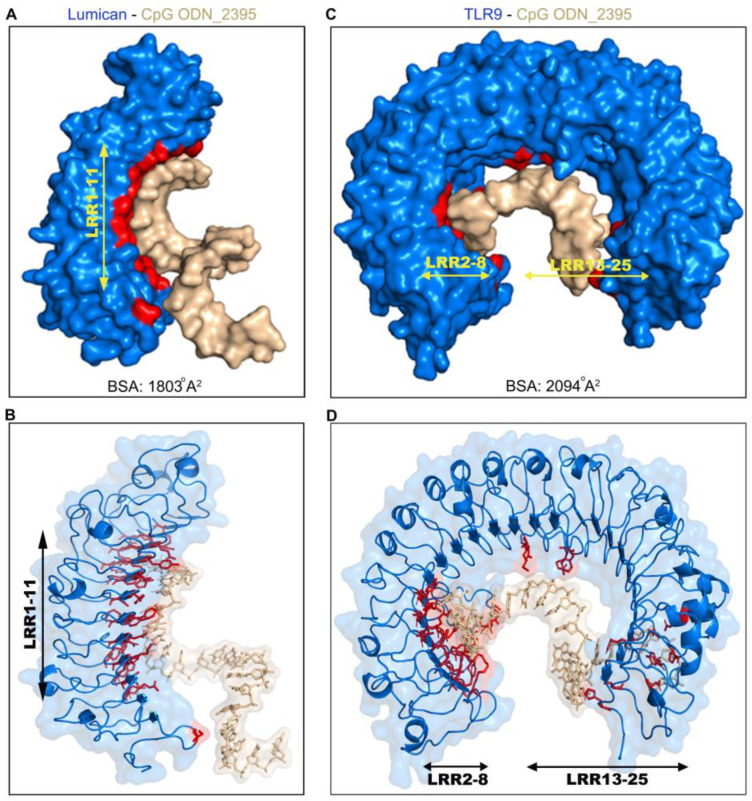
Modeling of CpG ODN_2395 bound Lumican. (**A**) Surface view of the lumican-CpG ODN_2395 complex shows the buried surface area (BSA) in red, lumican (blue), CpG ODN_2395 (golden). The combined BSA for Lumican-CpG DNA complex is 1803 Å^2^. Lumican residues contribute 42% (761 Å^2^) of the BSA, whereas CpG ODN_2395 contributes 58% (1042 Å^2^). (**B**) A ribbon 3D model showing all lumican residues (red) that interact with CpG ODN_2395 (golden). (**C**) Surface view of the TLR9-CpG DNA complex shows the buried surface area (BSA) in red, TLR9 (blue), and CpG ODN_2395 (golden). The combined BSA for the TLR9-CpG DNA complex is 2094 Å^2^. TLR9 residues contribute 42% (878 Å^2^) of the BSA. (**D**) A ribbon 3D model showing all TLR9 residues (red) that interact with the CpG ODN_2395 (golden).

**Figure 3 ijms-24-14990-f003:**
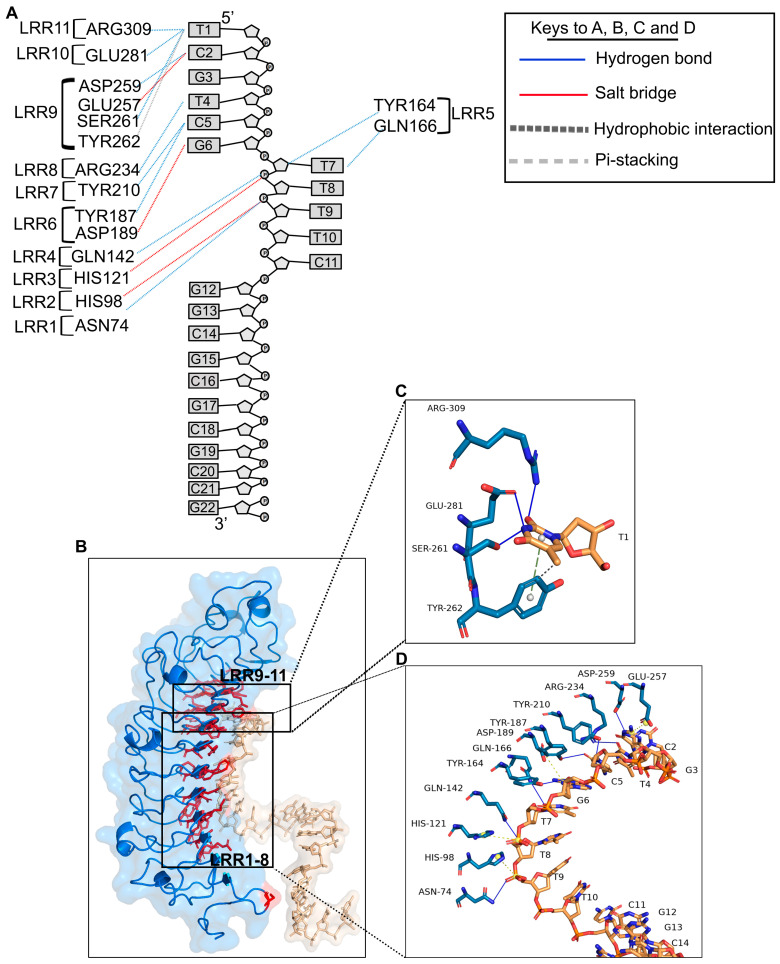
Close-up view of Lumican and CpG ODN_2395 interactions. (**A**) A schematic diagram showing lumican interacting with LRRs from lumican and nucleotides from CpG ODN_2395. The dotted lines in blue indicate H-bonds, red/yellow salt-bridges, and black pi-stacking interactions. (**B**) A 3D ribbon model showing all lumican residues (red) that interact with CpG ODN_2395 (golden). (**C**,**D**) A stick model showing the interaction of lumican with the first 5′ thymidine (T) base (**C**), and the next 8 bases of CpG ODN_2395 (**D**). The yellow ball shows the charge centers in CpG ODN_2395, and the grey ball indicates the aromatic charge centers of amino acids.

**Figure 4 ijms-24-14990-f004:**
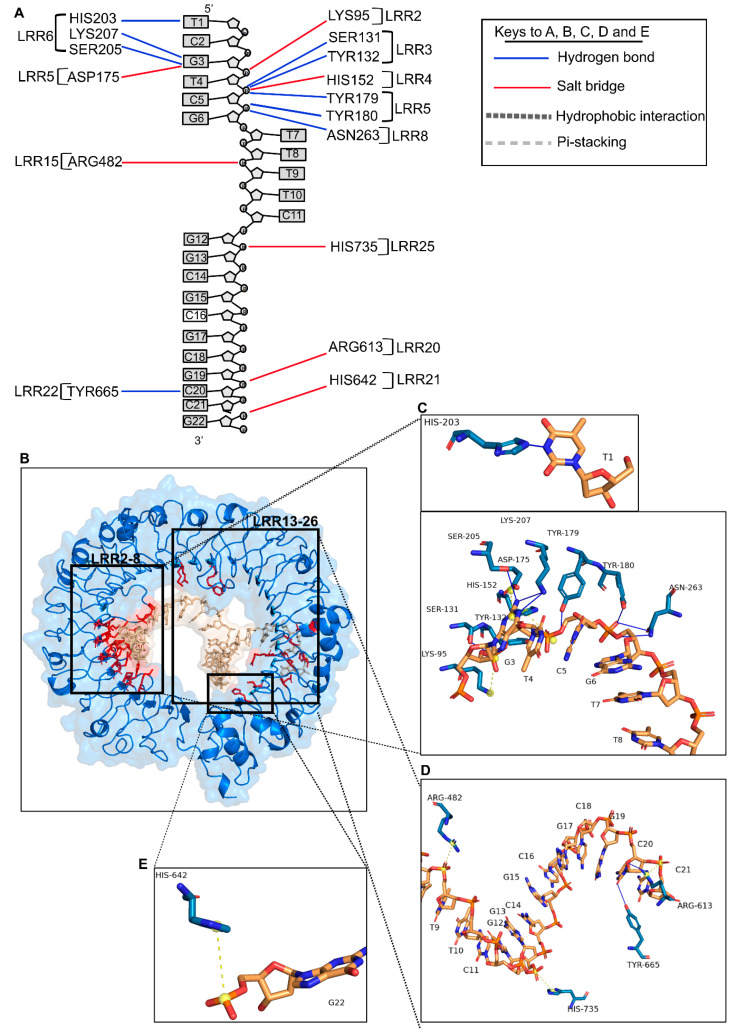
Close up view of TLR9 ECD and CpG ODN_2395 interactions. (**A**) A schematic diagram showing all TLR9 LRR and CpG ODN_2395 interactions. The lines in blue indicates H-bonds and red/yellow salt-bridges. (**B**) A 3D ribbon model showing all the TLR9 residues (red) that interact with the CpG ODN_2395 (golden). (**C**–**E**) Stick model indicates the interaction of TLR9 LRR2–8 with the bases or sugar phosphate backbone of the 5′-TCGTCG of CpG ODN_2395. (**C**) The LRR15, 20–22, and 25 interacting with phosphate backbone of the thymidine (T) at position 8, guanosine (G) at position 13, and cytosine (C) at position 20 of CpG ODN_2395, respectively. (**D**) LRR21 interacting with the phosphate backbone of guanosine (G) at position 22 of CpG ODN_2395. (**E**) The yellow ball shows the charge centers in CpG ODN_2395.

**Figure 5 ijms-24-14990-f005:**
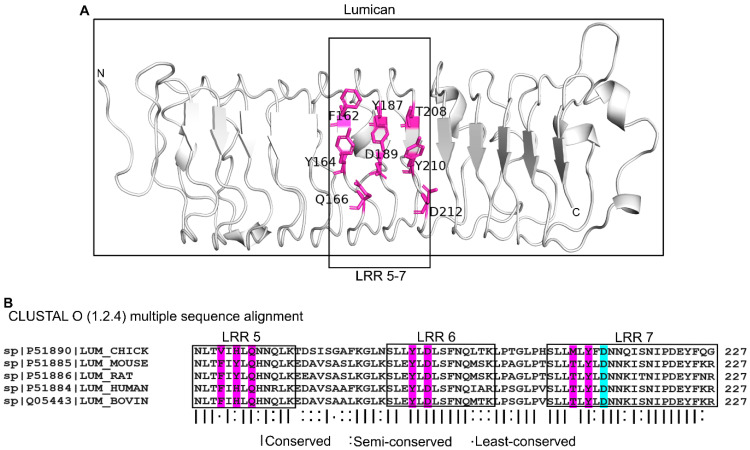
CpG DNA binding sites in LRR5–7 of lumican also known to bind collagen I. (**A**) CpG ODN_2395 binding residues in LRR5–7 of lumican. CpG ODN_2395 binding residues colored magenta. (**B**) Multiple sequence alignment of lumican LRR5–7 from different species. Interface residues are marked in magenta. Asp212 (D212) in cyan is the interface residue that interacts with type I collagen.

**Table 1 ijms-24-14990-t001:** Lumican—CpG ODN_2395 interacting amino acid residues.

Lum Interface	BSA	LRR	LRR Type
SER45	17%	NT	-
ARG73	22%	LRR1	T
ASN74	23%		
ILE95	34%	LRR2	T
ASP97	90%		
HIS98	42%		
LYS119	19%	LRR3	PS
HIS121	65%		
ASN123	72%		
TYR124	20%		
ASP140	59%	LRR4	SDS22
GLN142	96%		
PHE162	42%	LRR5	T
TYR164	99%		
GLN166	67%		
TYR187	94%	LRR6	S
ASP189	58%		
THE208	71%	LRR7	T
TYR210	93%		
ASP212	7%		
TYR232	59%	LRR8	SDS22
ARG234	71%		
SER236	84%		
HIS237	15%		
GLU257	77%	LRR9	SDS22
ASP259	94%		
SER261	100%		
TYR262	41%		
TYR279	86%	LRR10	Irregular
GLU281	94%		
VAL282	2%		
ARG309	37%	LRR11	Irregular

**Table 2 ijms-24-14990-t002:** TLR9—CpG ODN_2395 interacting amino acid residues.

TLR9 Interface	BSA	LRR	LRR Type
LYS95	29%	LRR2	Irregular
ASN129	57%	LRR3	S
SER131	100%		
TYR132	57%		
SER149	66%	-	-
SER151	85%	LRR4	T
HIS152	70%		
VAL171	48%	LRR5	Irregular
PHE173	79%		
ASP175	84%		
TYR179	63%		
TYR180	23%		
LYS181	9%		
HIS203	66%	LRR6	S
SER205	0%		
LYS207	67%		
TYR208	56%		
TYR224	52%	LRR7	T
LEU226	48%		
PRO262	17%	LRR8	Irregular
ASN263	48%		
GLN399	55%	LRR13	T
MET400	47%		
PHE467	37%	-	
ARG482	24%	LRR15	T
ARG613	36%	LRR20	Irregular
ASN640	52%	LRR21	T
HIS642	51%		
ARG662	23%	LRR22	T
TYR665	50%		
GLN689	15%	LRR23	SDS22
SER711	14%	LRR24	T
HIS735	20%	LRR25	T

**Table 3 ijms-24-14990-t003:** Different types of interaction between lumican and CpG ODN_2395. Blue: H-Bond Donor; Orange: H-Bond Acceptor.

Lum-CpG				
**Hydrogen bond**	**CpG ODN_2395**	**Atom**	**Lum**	**Atom**
	T1	N3	GLU281	OE1
		N3	SER261	OG
		O2	ARG309	NH2
	C2	N4	ASP259	OD2
	T4	O2	ARG234	NH1
	C5	N3	TYR210	OH
		O2	TYR187	OH
	G6	O4	TYR164	OH
	T7	O2	GLN166	NE2
	T8	O5	GLN142	NE2
	T9	OP1	ASN74	ND2
**Salt bridge**	**CpG ODN_2395**	**Charge Center**	**Lum**	**Charge Center**
	G3	Purine ring	GLU257	Carboxylic
	G6	Purine ring	ASP189	Carboxylic
	T8	P-backbone	HIS121	Imidazole
	T9	P-backbone	HIS98	Imidazole
**Van der Waals**	**CpG ODN_2395**	**Atom**	**Lum**	**Atom**
	T1	C7	TYR262	CE2
**Pi-stacking**	**CpG ODN_2395**	**Atom**	**Lum**	**Atom**
	T1	Pyrimidine ring	TYR262	Tyrosyl ring

**Table 4 ijms-24-14990-t004:** Different types of interaction between TLR9 and CpG ODN_2395. Blue: H-Bond Donor; Orange: H-Bond Acceptor.

TLR9-CpG				
**Hydrogen bond**	**CpG ODN_2395**	**Atom**	**TLR9**	**Atom**
	T1	N3	HIS203	NE2
	G3	N1	LYS207	NZ
		N2		NZ
		N2	SER205	OG
	T4	OP1	SER131	OG
		O3	TYR132	OH
	C5	O5	TYR179	OH
	G6	OP1	TYR180	OH
		OP1	ASN263	OD1
		O5		ND2
	C20	O2	TYR665	OH
		O4	ARG613	NH2
**Salt bridge**	**CpG ODN_2395**	**Charge Center**	**TLR9**	**Charge Center**
	G3	Purine ring	ASP175	Carboxylic
	T4	P-backbone	LYS95	Amine
	C5	P-backbone	HIS152	Imidazole
	T9	P-backbone	ARG482	Amine
	G13	P-backbone	HIS735	Imidazole
	C21	P-backbone	ARG613	Amine
	G22	P-backbone	HIS642	Imidazole

**Table 5 ijms-24-14990-t005:** Sequence coverage and sequence identity between the target sequence and the modeling sequence.

Lumican-CpG ODN_2395 Docking					
Template Molecule	PDB ID	Chain ID	Algin_length	Coverage (%)	Seq_identity (%)
TLR3 (Receptor)	7DA7	A	255 amino acids	76	31
dsRNA (Ligand)	7DA7	C	14 bases	64	79
**TLR9-CpG ODN_2395 Docking**					
**Template Molecule**	**PDB ID**	**Chain ID**	**Algin_length**	**Coverage (%)**	**Seq_identity (%)**
TLR3 (Receptor)	7DA7	A	733 amino acids	99	26
dsRNA (Ligand)	7DA7	C	14 bases	64	79

## Data Availability

The data and materials used to support the findings of this study are available from the corresponding authors (George Maiti or Shukti Chakravarti) upon request.

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
