# Peer review of "Three-Dimensional Modeling of CpG DNA Binding with Matrix Lumican Shows Leucine-Rich Repeat Motif Involvement as in TLR9-CpG DNA Interactions"

_ijms, 2023, doi:10.3390/ijms241914990_

Round 1
Reviewer 1 Report
This manuscript is written and presented very well according to the journal's requirements, and I would like to accept it in its present form.
Author Response
Thank you very much.
Reviewer 2 Report
The manuscript by Choi et. al. entitled with “3D Modeling of CpG DNA Binding with Matrix Lumican 2 Shows Leucine-Rich Repeat Motif Involvement as in 3 TLR9- CpG DNA Interactions” is an immature study that contains a few scientifically reliable findings but is not based on sufficient computational experiments, modellings, model-assessments to support the authors' claims. To this reviewer, it was a very hard time to estimate the feasibility of the two models, Lumican-CpG DNA and TLR9-CpG DNA complex models, because the authors seem not to correctly understand the theoretical basis of template-based molecular modelling, especially the case of the single stranded DNA.
The following is the list of the major issues and candidate of the points to be improved.
1. The building of the initial structure of CpG ODN ssDNA structure with its logic. The authors used B-dorm based CpG ssDNA structure instead of the PDB template, for example PDB 3WPE, 5ZLN or 5Y3M, all of which are TLR9-ssDNA complex structures. The authors mentioned that “Although TLR9 in complex with CpG ODN_1668 12mer exists in PDB database, the nucleotide sequence coverage was 35%, which only 8 nucleotide sequences were aligned out of 22 of CpG ODN_2395, and not sufficient for the template-based docking prediction.” (lines 404-406). However, this description was not scientifically appropriate. In case of the homology modelling of protein structure, the sequence identity is important parameter for the accuracy of the predicted model. However, in case of the ssDNA, the structure in solution is very flexible and not limited to the B-form structure, because B-form DNA structure is only observed at very limited solution condition with dsDNA. In PDB, there are more than 10 entries of the TLR8/TLR9 structures in complex with ssDNAs. All of these DNAs are very different from the theoretical B-form structure that the authors tried to used in this study. Instead, the authors can build the models using the template-based modeling method using these TLR9-bound ssDNAs with various aligned position combination, and then select the best scores by assessing molecular contacts in detail.
2. After success to build some more reliable model for TLR9-ssDNA, the similar approach can be used to build lumican with CpG ODN. However, lumican structure is just the AF2-prdicted model, and the sidechain orientations are not perfect. Thus the modeling with ssDNA must be carefully performed.
Author Response
Reviewer 2:
The manuscript by Choi et. al. entitled with “3D Modeling of CpG DNA Binding with Matrix Lumican 2 Shows Leucine-Rich Repeat Motif Involvement as in 3 TLR9- CpG DNA Interactions” is an immature study that contains a few scientifically reliable findings but is not based on sufficient computational experiments, modellings, model-assessments to support the authors' claims. To this reviewer, it was a very hard time to estimate the feasibility of the two models, Lumican-CpG DNA and TLR9-CpG DNA complex models, because the authors seem not to correctly understand the theoretical basis of template-based molecular modelling, especially the case of the single stranded DNA.
The following is the list of the major issues and candidate of the points to be improved.
Comment.1. The building of the initial structure of CpG ODN ssDNA structure with its logic. The authors used B-dorm based CpG ssDNA structure instead of the PDB template, for example PDB 3WPE, 5ZLN or 5Y3M, all of which are TLR9-ssDNA complex structures.The authors mentioned that “Although TLR9 in complex with CpG ODN_1668 12mer exists in PDB database, the nucleotide sequence coverage was 35%, which only 8 nucleotide sequences were aligned out of 22 of CpG ODN_2395, and not sufficient for the template-based docking prediction.” (lines 404-406). However, this description was not scientifically appropriate. In case of the homology modelling of protein structure, the sequence identity is important parameter for the accuracy of the predicted model. However in case of the ssDNA, the structure in solution is very flexible and not limited to the B-form structure, because B-form DNA structure is only observed at very limited solution condition with dsDNA. In PDB, there are more than 10 entries of the TLR8/TLR9 structures in complex with ssDNAs. All of these DNAs are very different from the theoretical B-form structure that the authors tried to used in this study. Instead, the authors can build the models using the template-based modeling method using these TLR9-bound ssDNAs with various aligned position combination, and then select the best scores by assessing molecular contacts in detail
Response: We appreciate the reviewer for pointing out this important fact. We have now addressed the first part of this comment in our revised manuscript (line 328-334). We understand that using B-form is a potential limitation of this study. However, a previous study by Temiz, et al, (New reference. 38) demonstrated by molecular dynamics simulation that unmethylated CpG dinucleotides tends to form the more flexible, right-handed canonical B-conformation under physiological conditions but methylation of the cytosine nucleotide favors the less mobile, left-handed, non-canonical Z-form. Therefore, it is logical to assume that under physiological conditions the CpG ODN_2395 containing unmethylated CpG dinucleotides will tend to form the right-handed canonical B-conformation that we have used in our simulation.
We clearly understand the second part of the reviewer concern and have explained it in the revised manuscript (line: 139-149 and 428-461) and added a new table.5 to show the similarity between the target sequence and the modeling sequence. To improve the docking credibility we initially tried to make HDOCK software choose TLR9-CpG complex in PDB database (3WPE or 3WPC) for the docking simulation, as you have suggested. However, the challenge in this case was the HDOCK software chooses the template based on maximum sequence coverage and maximum sequence identity for both ligand and receptor, and the threshold for sequence coverage is 50%. Indeed, the receptor TLR9 (PDB: 3WPC) in PDB has the best sequence coverage (100%) and identity (69%). However, the ligand CpG ODN_1668 12mer, which is the longest CpG DNA crystalized with TLR9 in the PDB database, gave only 6 aligned lengths with our CpG ODN_2395 (22 mer). The sequence coverage for the CpG DNA was 27% (6/22) which was too low to reach the HDOCK threshold of 50%. Therefore, HDOCK automatically excludes the template with sequence coverage < 50% from the candidate templates. Instead, HDOCK choose TLR3 complexed with the double stranded (ds) lncRNA Rmrp (PDB: 7DA7) as a common template for lumican and TLR9 docking with CpG ODN_2395 that gave sequence coverage greater than 50% for both ligand (CpG ODN_2395) and the receptors (lumican and TLR9) as in the table below. We also explained in the results section (lines 138-149).
|
Lumican-CpG_ODN_2395 Docking |
|||||
|
Template Molecule |
PDB ID |
Chain ID |
Algin_length |
Coverage (%) |
Seq_identity (%) |
|
TLR3 (Receptor) |
7DA7 |
A |
255 amino acids |
76 |
31 |
|
dsRNA (Ligand) |
7DA7 |
C |
14 bases |
64 |
79 |
|
TLR9-CpG_ODN_2395 Docking |
|||||
|
Template Molecule |
PDB ID |
Chain ID |
Algin_length |
Coverage (%) |
Seq_identity (%) |
|
TLR3 (Receptor) |
7DA7 |
A |
733 amino acids |
99 |
26 |
|
dsRNA (Ligand) |
7DA7 |
C |
14 bases |
64 |
79 |
The TLR3 and dsRNA interacts in similar way as TLR9 and CpG because TLR3/TLR9 binds to DNA/RNA by recognizing phosphate backbones. Both TLR3 and TLR9 interactions with its ligand involve N-terminus LRR1-13 as a major binding surface due to its positive charged surfaces that provides electrostatic interaction surface for ligand binding.
Comment.2. After success to build some more reliable model for TLR9-ssDNA, the similar approach can be used to build lumican with CpG ODN. However, lumican structure is just the AF2-prdicted model, and the sidechain orientations are not perfect. Thus the modeling with ssDNA must be carefully performed.
Response: Thank you for this suggestion. We have addressed this issue in the revised manuscript (line: 340-353). Indeed structural prediction is not a 100% representation of the actual protein structure but AlphaFold protein structure database is currently the best available tool that can predict highly accurate structures with a median backbone accuracy of 2.8 Å r.m.s.d.95 (95% confidence interval = 2.7-4.0 Å) and produces highly accurate side chains (Reference number. 28).
Reviewer 3 Report
In this manuscript, the authors present an in silico study of the interaction of CpG DNA and lumicam. I have no significant comments. The work is well done. The manuscript is written correctly, the methods are well-chosen, and the results are clearly presented. This study is interesting. I think the work deserves attention.
However, one method missing from this work is molecular dynamics. Molecular docking is ok, no my comments on this part, but in this kind of work, without in vitro or in vivo studies, this basic method is a bit too little. Molecular dynamics, combined with docking, provides much more information and provides more reliable conclusions. Why didn't the authors use MD simulation?
Authors should carefully check the manuscript for editing. E.g. line 99 (2.1 3D not 2.1.3 D, line 336(wwPDB?) or the format of citations and literature references are not according to the template of IJMS.
Author Response
Reviewer 3:
In this manuscript, the authors present an in silico study of the interaction of CpG DNA and lumican. I have no significant comments. The work is well done. The manuscript is written correctly, the methods are well-chosen, and the results are clearly presented. This study is interesting. I think the work deserves attention.
Comment.1. However, one method missing from this work is molecular dynamics. Molecular docking is ok, no my comments on this part, but in this kind of work, without in vitro or in vivo studies, this basic method is a bit too little. Molecular dynamics, combined with docking, provides much more information and provides more reliable conclusions. Why didn't the authors use MD simulation?
Response: Thank you for your suggestions. We have now addressed this concern in the revised manuscript (line: 336-341) as a potential limitation of our study. As you pointed out, it is important to perform molecular dynamics simulation to investigate binding stability and conformational changes using RMSD and RFSD values generated from MD. In this rigid docking analysis study, we have included buried surface area (BSA) calculations and bonding profiles to indicate the binding stability and affinity between the ligand and receptors. We have now included a new reference (Reference number-43) to support this point (line: 519-521). However we understand that MD analysis is the limitation of this study and this is our next step, with additional in vivo experiments.
Comment.2. Authors should carefully check the manuscript for editing. E.g. line 99 (2.1 3D not 2.1.3 D, line 336(wwPDB?) or the format of citations and literature references are not according to the template of IJMS.
Response: We have now addressed these minor issues in the revised manuscript.
Reviewer 4 Report
In the manuscript titled:” 3D Modeling of CpG DNA Binding with Matrix Lumican Shows Leucine-Rich Repeat Motif Involvement as in TLR9- CpG DNA Interactions” the authors report a computational study involving lumican interactions with CpG-DNA. The results are interesting to find novel molecular targets for modulating TLR9 which is involved in inflammation and in autoimmunity. This is a preliminary study, because it is based only on docking, which takes into account rigid interactions between macromolecules in vacuo. The further step should be a study by molecular dynamics.
Comments:
11. The references in the text should be reported in square bracket following the guide to the authors.
22. In figures containing the electrostatic potential surface, it is not clearly explained if the positively charged and the negatively charged correspond to neutral or protonated/deprotonated amino acid forms.
33. In Table 3, please reduce the font to visualize completely the words “Van der Waals” and “pi-stacking”.
44. At rows 365, 367, 377 and 378 please report the unit of measure for RMSD, as well as at row 384 and 388 for grid spacing.
55. At row 323, please insert a reference for what is written in the sentence.
66. In paragraph 4.4 it is not clear if CpG ODN_2395 structure after built by Discovery Studio was minimized and which force field was used?
77. Due to the complexity of the results and discussion, please insert the conclusions paragraph, reporting synthetically the most important obtained results.
The manuscript can be accepted after minor revision
Author Response
Reviewer 4:
In the manuscript titled:” 3D Modeling of CpG DNA Binding with Matrix Lumican Shows Leucine-Rich Repeat Motif Involvement as in TLR9- CpG DNA Interactions” the authors report a computational study involving lumican interactions with CpG-DNA. The results are interesting to find novel molecular targets for modulating TLR9 which is involved in inflammation and in autoimmunity. This is a preliminary study, because it is based only on docking, which takes into account rigid interactions between macromolecules in vacuo. The further step should be a study by molecular dynamics.
Comment.1. The references in the text should be reported in square bracket following the guide to the authors.
Response: We have now addressed this issue in the revised manuscript.
Comment.2. In figures containing the electrostatic potential surface, it is not clearly explained if the positively charged and the negatively charged correspond to neutral or protonated/deprotonated amino acid forms.
Response: Thank you for pointing this out. An understanding of electrostatic interactions is important in the study of receptor-ligand interactions. The APBS electrostatics software only allows us to visualize the distribution of acidic and basic amino acid residues on the 3D surface of proteins. However the software does not provide any details about the protonation or deprotonation status of the individual residues. We have stated this issue in the revised manuscript (line: 420-423).
Comment.3. In Table 3, please reduce the font to visualize completely the words “Van der Waals” and “pi-stacking”.
Response: We have now addressed this issue in the revised manuscript.
Comment.4. At rows 365, 367, 377 and 378 please report the unit of measure for RMSD, as well as at row 384 and 388 for grid spacing.
Response: We have now reported the appropriate units in the revised manuscript.
Comment.5. At row 323, please insert a reference for what is written in the sentence.
Response: We have inserted the appropriate reference for this statement in the revised manuscript.
Comment.6. In paragraph 4.4 it is not clear if CpG ODN_2395 structure after built by Discovery Studio was minimized and which force field was used?
Response: We have now addressed this point in the revised manuscript (line: 421-422). The CpG ODN_2395 structure was built by using the B DNA conformation setting available in the Discovery Studio without minimizing the structure. The following settings was used: right handed, 10 base pair per turn, 3.4 Å vertical rise per base pair, helical diameter- 19 Å.
Comment.7. Due to the complexity of the results and discussion, please insert the conclusions paragraph, reporting synthetically the most important obtained results.
Response: We have now added a short conclusion for the readers in the revised manuscript at the end of the discussion section.
Round 2
Reviewer 2 Report
The revised version of manuscript by Choi et. al. entitled with “3D Modeling of CpG DNA Binding with Matrix Lumican Shows Leucine-Rich Repeat Motif Involvement as in 3 TLR9- CpG DNA Interactions” seems still an immature study that contains a few scientifically reliable findings but is not based on right-use of computational experiments, modellings, model-assessments to support the authors' claims. In this revised version, the authors referred the new reference [38], “The role of methylation in the intrinsic dynamics of B- 607 and Z-DNA. PLoS One 2012, 7:e35558”, as the evidence that the ssDNA CpG ODN tends to form a B-form structure in solution before binding with TLR9. However, the reference is just the 100 ns MD simulation, and even in a such very short time MD simulation, the initial B-form structure largely distorted. Thus, this reviewer thinks the reference is NOT appropriate to be referred as the theoretical basis that the authors can use the B-form-originated starting model for this docking study.
In addition, computational research of ssDNA of a given nucleotide sequence seems to have a long history, especially for molecular dynamics simulation with explicit solvent systems. Most of these studies with some experiments, for example, SAXS experiments, showed that ssDNA adopts into an extended conformation rather than a B-form rigid structure. The authors claims completely neglect these knowledges and efforts of computational scientists for ssDNA, and the mechanisms of molecular recognition by other biological molecules.
One possible solution is to use a short (100ns) MD simulation of ssDNA CpG ODN before docked with TLR9, and the results of docking study may become some more realistic and reliable. For this additional pretreatment of ssDNA, probably AMBER force filed is the best candidate (for example, see PMID: 33434436).
The additional authors’ effort to provide the new Table 5 is commended. However, this is only the explanation why the authors did not use the template-based modelling approach from the existing TLR8/9 – ssDNA complexes in PDB, and the theoretical basis and its scientific validity is so limited.
Author Response
Reviewer#2
The revised version of manuscript by Choi et. al. entitled with “3D Modeling of CpG DNA Binding with Matrix Lumican Shows Leucine-Rich Repeat Motif Involvement as in 3 TLR9- CpG DNA Interactions” seems still an immature study that contains a few scientifically reliable findings but is not based on right-use of computational experiments, modellings, model-assessments to support the authors' claims. In this revised version, the authors referred the new reference [38], “The role of methylation in the intrinsic dynamics of B- 607 and Z-DNA. PLoS One 2012, 7:e35558”, as the evidence that the ssDNA CpG ODN tends to form a B-form structure in solution before binding with TLR9. However, the reference is just the 100 ns MD simulation, and even in a such very short time MD simulation, the initial B-form structure largely distorted. Thus, this reviewer thinks the reference is NOT appropriate to be referred as the theoretical basis that the authors can use the B-form-originated starting model for this docking study.
In addition, computational research of ssDNA of a given nucleotide sequence seems to have a long history, especially for molecular dynamics simulation with explicit solvent systems. Most of these studies with some experiments, for example, SAXS experiments, showed that ssDNA adopts into an extended conformation rather than a B-form rigid structure. The authors claims completely neglect these knowledges and efforts of computational scientists for ssDNA, and the mechanisms of molecular recognition by other biological molecules.
One possible solution is to use a short (100ns) MD simulation of ssDNA CpG ODN before docked with TLR9, and the results of docking study may become some more realistic and reliable. For this additional pretreatment of ssDNA, probably AMBER force filed is the best candidate (for example, see PMID: 33434436).
The additional authors’ effort to provide the new Table 5 is commended. However, this is only the explanation why the authors did not use the template-based modelling approach from the existing TLR8/9 – ssDNA complexes in PDB, and the theoretical basis and its scientific validity is so limited.
Response: We understand the reviewer concern. Now we have added a separate paragraph in the discussion (highlighted in yellow) to address reviewer concern mentioning 12 new appropriate references to explain the limitations of our study and the perspectives for further study.